# Behavioral Evolution of *Drosophila*: Unraveling the Circuit Basis

**DOI:** 10.3390/genes11020157

**Published:** 2020-02-01

**Authors:** Kosei Sato, Ryoya Tanaka, Yuki Ishikawa, Daisuke Yamamoto

**Affiliations:** 1Neuro-Network Evolution Project, Advanced ICT Research Institute, National Institute of Information and Communications Technology, Iwaoka, Nishi-ku, Kobe, Hyogo 651-2492, Japan; 2Division of Biological Science, Graduate School of Science, Nagoya University, Furo, Chikusa-ku, Nagoya, Aichi 464-8602, Japan

**Keywords:** the *fruitless* gene, species-specific behavior, identified neurons, nuptial gift, pheromone perception, courtship songs

## Abstract

Behavior is a readout of neural function. Therefore, any difference in behavior among different species is, in theory, an outcome of interspecies diversification in the structure and/or function of the nervous system. However, the neural diversity underlying the species-specificity in behavioral traits and its genetic basis have been poorly understood. In this article, we discuss potential neural substrates for species differences in the courtship pulse song frequency and mating partner choice in the *Drosophila melanogaster* subgroup. We also discuss possible neurogenetic mechanisms whereby a novel behavioral repertoire emerges based on the study of nuptial gift transfer, a trait unique to *D. subobscura* in the genus *Drosophila*. We found that the conserved central circuit composed primarily of *fruitless*-expressing neurons (the *fru*-circuit) serves for the execution of courtship behavior, whereas the sensory pathways impinging onto the *fru*-circuit or the motor pathways downstream of the *fru*-circuit are susceptible to changes associated with behavioral species differences.

## 1. Introduction

Every animal species displays a unique behavioral pattern, which often allows us to distinguish one species from others at a glance, even when the morphological characteristics of the species are highly similar. Such species-specific behavioral patterns may in turn lead to morphological changes as in Darwin’s finches, where diversified feeding preference likely drove the evolution of a variety of beak shapes among closely related species with distinct feeding habits [1]. As Konrad Lorenz inferred from the latent down-up courtship action in some species of goose, species-specific gain or loss of a discrete behavioral action could arise from a subtle change in gene expression [2], yet we know very little about the molecular genetic basis of this process. In *Drosophila*, cross-species comparative studies on mating behavior have highlighted remarkable diversities in courtship rituals among not only species belonging to distant clades but also those within the same species complex [3]. The best-studied element in the ritual is the male courtship song generated by wing vibration, the temporal structure of which varies depending on species [4]. However, some species produce songs by vibrating body parts other than wings [5] or by moving the substrates they sit on [6,7]. In yet another species, both females and males coordinately sing [4,8,9]; such coordinated singing typically occurs before copulation (premounting song), although there are also fly species in which individuals sing after copulation with or without a premounting song [10]. Therefore, both qualitative and quantitative differences in singing behavior account for a rich variety in courtship rituals. 

Behavior represents an ensemble of motor outputs produced by dedicated neural circuits. Thus, any change in a behavioral pattern is a consequence of alterations in the structure (wiring) and/or function (physiology) of the neural circuit underlying the behavior [11]. Therefore, a deep understanding of the circuit basis for a given behavior is a prerequisite for gaining evolutionary insights into the mechanistic underpinnings of behavioral diversification. *Drosophila melanogaster* offers an unparalleled model system for untangling the complex wiring of neurons in the brain for anatomical and functional tracing of a circuit; its suitability is due to the availability of techniques and tools for labeling and manipulating single neurons, and to the large body of accumulated genetic resources specific to this species [12,13,14,15]. In particular, the architecture of circuits for courtship behavior has been revealed in some depth in this species, partly due to the discovery of the *fruitless* (*fru*) gene, which orchestrates the courtship circuit formation as a master regulator transcriptional factor gene [16,17,18,19]. Comparative studies of the *fru*-labeled courtship circuit have opened an avenue for exploring the latent neural evolution underlying diversification in courtship rituals of *Drosophila* species. 

In this article, we focused on three aspects of courtship behavior: first, the neural basis for species differences in song characteristics among the members of the *D. melanogaster* subgroup; second, the neural basis for conspecific preference in mating partner choice between *D. melanogaster* and its sibling species *D. simulans*; third, the neural basis for nuptial gift transfer as a courtship ritual evolved exclusively in *D. subobscura*, a distant relative of *D. melanogaster*. Studies on these three traits represent exceptional cases in which neuronal groups responsible for the evolutionary remodeling of courtship rituals have been elucidated in some detail. Additionally, the species difference in these traits shows different levels of diversification, ranging from a quantitative difference in sensorimotor characteristics for a behavioral element shared by the sibling species to a qualitative difference due to the recruitment of a new repertoire to the courtship ritual only in select species. 

## 2. *D. melanogaster* Male Courtship Ritual

*D. melanogaster* male mating behavior comprises several discrete motor acts, including orientation of the body axis toward a target female, chasing her from behind, tapping her abdomen, circling around her, wing extension and vibration for courtship song generation, approaching and licking her genitalia, attempting to mount her with abdominal bending, mounting, and copulation [20,21]. Among these, courtship song has been intensively studied as it is a species-specific trait that is amenable to rigorous quantitative comparisons. Traditionally, the courtship song in *D. melanogaster* has been categorized into two types, i.e., pulse song and sine song: the pulse song is composed of a series of tone pulses with an average inter-pulse interval (IPI) of ~35 ms, whereas the sine song represents oscillatory tones with a mean peak frequency of ~230 Hz at room temperature [22] (Figure 1A). Both pulse and sine songs promote female acceptance of mating [22,23], and pulse song also elicits courtship activities in other males, presumably by acting as an anticipatory cue for the presence of the female [24,25]. The pulse song IPI varies from species to species [26,27], and flies respond to artificial pulse songs that have characteristics of conspecific songs [28], including the pulse song IPI [29]. Thus, the IPI of pulse song seems to convey the information of species identity of the singer. A recent intense analysis of courtship songs in *D. melanogaster* led to the distinction of two types of pulse song, fast pulse song (P_fast_) and slow pulse song (P_slow_), based on the differences in their time course and shape [30]. A male generates P_fast_ by fully extending a wing (~60°) when he is apart from a target female. In contrast, the male more often generates P_slow_ by weakly extending a wing (~20°) upon approaching and coming into proximity of the female [30].

## 3. Circuit Basis for Courtship Behavior in *D. melanogaster*

The P1 cluster, a male-specific interneuron group capable of initiating male courtship behavior (Figure 1B), was identified by clonal sexual transformation of these neurons in the brain of females: more than 80% of “females” carrying this 20 cell cluster (sexually mosaic flies harboring the male-specific P1 cluster) in their brain performed some male courtship actions to court other females, including orientation, following, and unilateral wing extension and vibration [31]. P1 neurons express *fru* and the other sex-determinant transcription factor gene, *doublesex* (*dsx*). P1 neurons are confined within the brain, densely innervating the lateral protocerebrum (lpr) of both hemispheres by forming a thick midline fiber bundle called the “arch” [31,32]. The ipsilateral branches contribute to the “ring” and “lateral crescent”, two prominent neurite fascicles, in which multiple *fru*-positive neurons intermingle [32]. A subsequent mosaic analysis with males used a warmth-sensitive dTrpA1 channel minigene to selectively activate, by heating, clonal neurons that express this minigene. This attempt successfully identified P1 neurons, along with an additional cluster called P2b (Figure 1B), as cells that can induce courtship in a solitary male upon an increase in ambient temperature beyond 30 °C [33]. The second experiment using males mosaic for dTrpA1 did identify P2b as a neural cluster with the ability to induce courtship behavior, but the first experiment with sexually mosaic “females” did not so identify P2b. What was the reason for this discrepancy? In the first experiment, natural stimuli derived from a target female ultimately activated neurons that make the decision to court a target. In the second experiment, in contrast, an experimenter activated the neurons by artificially increasing the temperature. We infer that P2b neurons are unable to make the decision to court, but rather must be activated by other neurons, e.g., P1 neurons, to trigger courtship behavior. Indeed, dendritic arbors of P2b neurons are superimposed on branches of P1 neurons in the lpr, suggesting that synaptic contacts exist between them. P2b neurons extend long, descending axons down to the thoracic ganglia, where presumptive motor pattern generators for courtship behavior exist. It is therefore likely that P1 neurons make the decision to court and the decision is conveyed to the motor center via P2b neurons. 

Another study achieved activation of subsets of *fru*-expressing neurons by an intersectional approach with the combinatorial use of *fru*-neuron-specific flippase (*fru^FLP^*), flappable *dTrpA1* (*UAS>STOP>dTrpA1*), and GAL4 driver transgenes [34]. This study also identified P1 and a pair of descending neurons named pIP10, likely a member of P2b, as neurons that can initiate male courtship singing (Figure 1B). Other studies identified the *dsx*-positive pC2l cluster [35] and the *fru*-positive aSP22 descending neurons [36] as neurons activating licking and attempted copulation, two later-phase components of mating behavior, in addition to following pursuits and singing. 

Von Philipsborn [34] also identified a few intrinsic neurons important for song patterning in the thoracic ganglia (Figure 1B). For example, dTrpA1-mediated activation of vPR6 interneurons shortened the IPIs of pulse songs, and vMS11 interneurons induced, upon artificial activation, unilateral wing extension with no wing vibration, whereas, when silenced, pulse song (normally with a single peak, i.e., monocyclic pulse) was distorted to have multiple peaks (polycyclic pulse). A subsequent study employed Ca^2+^ imaging of muscle activities and neural silencing during courtship to characterize motor units composed of several motoneurons and their target muscles that contribute to song generation [37]. According to this work, overlapping motor units are activated during pulse and sine songs, yet pulse song production requires recruitments of additional motor units (for an alternative view, see [38]). Thus, the entire motor pathway for the generation of courtship song, from the courtship decision-making center to the muscles that mechanically move individual body parts for courtship actions, has been described in *D. melanogaster* (Figure 1C,D). These findings in *D. melanogaster* provide a solid basis for unraveling precisely where in the circuit homologous among different *Drosophila* species the divergence exists that may account for species differences in courtship behavior.

## 4. Neural Basis for Species-Specific Song Characteristics

Although the overall pattern of courtship behaviors is conserved among the members of the *D. melanogaster* subgroup, the song characteristics vary widely. Among nine species examined, three species lacked sine song, and pulse song was absent from one species [39]. Some species had two forms of pulse and/or sine songs, whereas others had only one type of pulse and sine songs [39]. The amplitude (loudness) of songs and the shape and IPI of pulse songs are distinctly different from species to species. Ding et al. [39] attempted a comparative approach of the *fru*-labeled neural pathway to deduce which neural component is responsible for song differences between *D. melanogaster* and *D. yakuba*. *D. yakuba* males generate two forms of pulse song, called clack song and thud song, but do not generate sine song. The clack song is produced by simultaneous vibration of two wings and contains higher carrier frequency components over 400 Hz, in contrast to the thud song, which is generated by unilateral wing vibration and has a carrier frequency of ~200 Hz. Clack song is greater in amplitude than thud song, and more often generated when the target female is distantly located from the courting male [39]. This parallels the case of *D. melanogaster*, which also has two forms of pulse song: P_fast_ is louder than P_slow_ and is typically produced when the male is courting at a distance [30]. In *D. melanogaster*, selective activation of pIP10 neurons via a split-GAL4 intersection primarily evoked P_fast_, and much less often P_slow_ and sine songs [30]. Ding et al. [39] introduced split-GAL4 constructs developed for *D. melanogaster* into the *D. yakuba* genome, and succeeded in selectively visualizing and manipulating a pIP10 homolog in this species. It turned out that while pIP10 in *D. yakuba* shared structural, electrical, and neurochemical properties with the *D. melanogaster* counterpart, optogenetic activation of *D. yakuba* pIP10 initiated predominantly (but not exclusively) clack song. This observation led them to propose that the command fiber role of pIP10 is conserved in two species: pIP10 preferentially triggers the large amplitude song type with a faster time course for long distance calls in both *D. melanogaster* and *D. yakuba*, i.e., pulse song (P_fast_ in particular) and clack song, respectively [39]. However, the structure and frequency of induced songs differ between the two species, suggesting that the pIP10 neuron connects with the motor center that specifies quantitative song characteristics inherent to each species. Which neurons in the motor center are responsible for the differences in quantitative parameters of courtship songs? This critical question remains unsolved.

## 5. Genetic Basis for Species-Specificity in Courtship Song Characteristics

Extensive searches for genes that potentially drove the song diversification among closely related species of *Drosophila* have been carried out by interspecific quantitative trait loci (QTL) analysis [40]. For example, the *maleless* (*mle*), *fru*, and *croaker* (*cro*) genes were implicated in the IPI difference between *D. simulans* and *D. sechellia* in the *D. melanogaster* species subgroup [41]. The Mle protein is an RNA helicase essential for doubling the transcriptional activity of the sole X-chromosome present in males so as to equalize the amounts of X-derived mRNAs in males with that in females with two X chromosomes, i.e., gene dosage compensation [42]. The *no action potential* (*nap*) allele of *mle* fails in nerve conduction at a restrictive temperature [43], accompanying prolongation of pulse song IPIs [44]. These behavioral phenotypes possibly result from disrupted splicing of the *paralytic* (*para*) Na^+^ channel gene [45], which may reduce fidelity in action potential firing in the courtship song circuit. The *cro* gene encodes the calmodulin-binding transcription factor (Camta) and displays multiple defects in mating behavior, including prolongation of pulse song IPIs [46,47]. The *fru* gene encodes a series of male-specific FruM transcription factor proteins that function as master regulatorw of male courtship circuit formation [19]. *fru* mutations affect a range of courtship behavioral elements, likely because the male courtship circuit forms through mutual connections among *fru*-expressing neurons. Indeed, *fru* mutant males devoid of functional FruM almost never sing courtship songs, and hypomorphic *fru* mutant males sing aberrant songs [48]. Nonetheless, cross-species transplantation of the intact genomic *fru* locus including the presumptive regulatory region as well as all exons and introns using BAC recombineering and P[acman] transgenesis did not affect species-specific song patterns: the host *D. melanogaster* male which received a non-*melanogaster fru* locus continued to produce courtship song characteristic of *D. melanogaster*, regardless of whether he was a mutant for *fru* and/or *dsx* [49]. This observation is not surprising, however, in view of the fact that even females can generate courtship-song-like sounds if *fru*-positive neurons in the ventral nerve cord [50] or the pC1 cluster, a female homolog of the male P1 cluster in the brain [51], are artificially activated. Because females have no FruM protein, these observations could suggest that FruM protein might be dispensable for *fru*-positive neurons to form a fundamental circuit required for courtship song generation. FruM might be important just for optimizing the male circuit to court conspecific females. There is an alternative explanation for these apparently conflicting observations, however. In the cross-species *fru*-locus transplantation experiment by Cande [49], males were required to sense female-derived stimuli to initiate courtship, whereas in the experiment where females were forced to sing, neurons were directly activated without any sensory involvement [50,51]. It would be of interest to examine, using Cande’s flies, whether the song characteristics would differ from those of the host song if the neurons were directly stimulated via the heterospecific *fru* promoter introduced into the host genome in the absence of a target female. Natural sensory stimuli could preferentially activate the endogenous *fru* song motor circuit, and the coexisting heterologous *fru* song motor circuit formed by an exogenous *fru* promoter transplanted from a different species could remain silent. In this case, the manipulated fly would produce a song inherent to the host species and not produce a song of the donor species. 

A more recent study attempted to unravel the genetic basis for the quantitative difference in sine song between *D. simulans* and *D. mauritiana* [52]. QTL analysis and multiplexed shotgun genotyping of F1 hybrids and their backcrossed progeny with the *D. simulans* strain *sim5* and the *D. mauritiana* strain *mau29* were performed. The result pointed to a retrovirus insertion into the *slowpoke* (*slo*) locus of *sim5*, which appeared to impair splicing of *slo* transcripts and ultimately reduce the carrier frequency of sine song (as much as ~10Hz) in *sim5* relative to that in *mau29* [52]. The *slo* locus encodes a Ca^2+^ activated K^+^ channel also known as big K or K^Ca^1.1 [53], the loss of which results in hyperexcitability of nerve and muscle [54]. However, subsequent examination of sine-song carrier frequencies in 12 independent isolates for each of *D. simulans* and *D. mauritiana* revealed that the carrier frequency is highly variable among individuals within both species, and the difference in this trait found between *sim5* and *mau29* fell into the range of intraspecies variations [52]. This study provided insights into the possible source of song variations among fly populations, yet the identification of genes responsible for species differences in song characteristics remains a major challenge for future studies.

## 6. Conspecific Partner Preference Involves Species-Specific Neural Connections

*D. simulans* males prefer *D. simulans* females over *D. melanogaster* females as mating partners (Figure 2A). This conspecific preference by *D. simulans* males is in part due to the *melanogaster*-female-specific pheromone 7,11-heptacosadiene (7,11-HD) (Figure 2B), which acts as an aphrodisiac for *D. melanogaster* males [55] and at the same time as a repellant for *D. simulans* males [56]. 7,11-HD is a hydrocarbon component of cuticle and is perceived by the tarsal gustatory receptor neurons that express *pickpocket 23* (*ppk23*) and *ppk25*, two ENaC channel genes (Figure 2C) [57,58]. Kohatsu [33] demonstrated that the courtship decision-making P1 neurons in a test *melanogaster* male tethered on a treadmill were excited when he touched the abdomen of a conspecific female but not a conspecific male with his tarsus. Therefore, the contact-chemical pheromone information received at the tarsi is sent to P1 neurons. Kohatsu and Yamamoto [35] later showed that P1 neurons were recurrently activated throughout courtship pursuits by the male. 

Subsequent studies showed that the 7,11-HD-induced excitatory input arrives at P1 across two synapses through an intervening ascending interneuron, i.e., *fru*-positive vAB3 [59] or *fru*-negative PPN1 [60] (Figure 2C). Notably, these studies identified an additional route for the 7,11-HD information to P1 neurons. This second pathway was mediated by *fru*-positive mAL interneurons [59,60], which were previously shown to be GABAergic [61] (Figure 2C). mAL neurons are also known to be postsynaptic to *Gr32a* gustatory neurons that respond to another pheromone substance, 7-tricosene (7-T), in males but presumably not females, because sexually dimorphic mAL dendritic trees [62] seem to contact the *G32a* sensory axon terminals only in males [61]. 7-T is more abundant in males than females of *D. melanogaster,* and in fact 7-T acts as a repellant for males [63]. In contrast, both females and males of *D. simulans* contain a large amount of 7-T. In line with their GABAergic nature, mAL neurons exert an inhibitory effect on P1 neurons when artificially activated or activated by an afferent pathway stimulation. Thus, sensing 7,11-HD at the periphery not only excited but also inhibited the courtship decision-making P1 neurons, although the ultimate outcome of the convergence of these two seemingly conflicting inputs was to induce excitation in P1 under the experimental conditions of these studies [59,60] (Figure 2C). mAL neurons might function as a gain adjuster for P1 activation, since mAL neurons receive both attractant and repellant stimuli from potential mating partners and thus are likely capable of evaluating the nature of the target. Notably, olfactory inputs initiated at the *Or67d*-expressing neurons that are responsive to the inhibitory male pheromone *cis*-Vaccenyl acetate cVA [64] are similarly sent through parallel excitatory and inhibitory pathways to P1 [59]. In addition to the processed sensory inputs from pheromone receptor neurons, internal states affect P1 activities via excitatory dopaminergic [65] and inhibitory NPFergic modulatory signals [66,67]: dopamine attenuates mAL-mediated inhibition to elevate P1 activities [68]. 

If the opposite responses to 7,11-HD are indeed the primary cause for the conspecific preference in mating partner selection in *D. melanogaster* and *D. simulans*, then the pathway connecting *ppk23/25*-expressing pheromone receptor neurons and P1 neurons is the most promising site in which to search for interspecies differences in the neural substrate for mating behavior. Indeed, Seeholzer [69] first confirmed, by generating *ppk23* mutants in *D. simulans*, that males of this species lost the conspecific preference for a mate when they lost functional *ppk23*. They then introduced a P1-specific GAL4 construct commonly used in *D. melanogaster* into the *D. simulans* genome, making it possible to record P1 Ca^2+^ activities and to optogenetically stimulate P1 in *D. simulans*. When P1 neurons of *D. simulans* were irradiated with strong light stimuli, these males robustly displayed indiscriminative courtship toward both *D. simulans* and *D. melanogaster* females, as was expected if *D. simulans* P1 neurons had a courtship-initiating ability as do *D. melanogaster* P1 neurons [69]. However, neither *D. simulans* females nor *D. melanogaster* females induced increases in Ca^2+^ in *D. simulans* male P1 neurons when the test *D. simulans* male touched the female abdomen [69], in sharp contrast to the robust Ca^2+^ increases in *D. melanogaster* P1 neurons observed when the test *melanogaster* male touched a conspecific female abdomen [33,69]. Notably, peripheral *ppk23* pheromone receptor neurons responded to 7,11-HD and induced activities in vAB3 even in *D. simulans* [69]. Based on these observations, Seeholzer [69] suggested that *D. simulans* males do not rely on pheromones in initiating courtship, although *ppk23*-responsive sensory neurons likely form functional synapses on the vAB3 neuron (see also Reference [70]). Although direct stimulation of vAB3 neurons by acetylcholine iontophoresis induced clear Ca^2+^ rises in these neurons in both *D. simulans* and *D. melanogaster* males, little response was elicited in P1 in *D. simulans*, unlike in *D. melanogaster*, in which obvious Ca^2+^ increases were evoked upon vAB3 stimulation, revealing a remarkable decline in the ability of vAB3 to activate P1 in *D. simulans* compared to *D. melanogaster* [69]. Seeholzer [69] proposed a model in which inhibition of P1 neuron activities imposed by input from mAL neurons are stronger in *D. simulans* than in *D. melanogaster*, such that vAB3 neurons are unable to trigger courtship even when vAB3 neurons become active by incoming 7,11-HD-induced afferent signals (Figure 2D). 

To our knowledge, the present work is the sole successful case of a species-difference in behavioral responses being traced down to the functional divergence of a neural circuit at the identified neuron level. It remains an open question how the mAL neurons of *D. simulans* achieve more efficient inhibition of P1 neuron activities than their *D. melanogaster* counterparts. Unraveling the molecular entities contributing to the species difference in vAB3-P1 signaling efficacy may provide clues to the identity of the causative genes that drove neural divergence in the mate choice circuits between *D. simulans* and *D. melanogaster*.

## 7. Nuptial Gift Transfer as a Novel Component of the Courtship Ritual

In the preceding sections, we overviewed studies exploring the neural divergence underlying quantitative differences in mating behavior among members of the *D. melanogaster* subgroup in the genus *Drosophila*. Here, we consider the neural bases for divergent behavior among *Drosophila* species which are distantly related to each other and thus unable to produce interspecies hybrids. Since an ancestral species of the *obscura* species group and the *melanogaster* group diverged 25 million years ago, distinct mating behavior has evolved in these two clades. *D. subobscura* is particularly interesting, because nuptial gift transfer in this species is indispensable for acceptance of a male by a female as a mating partner [71,72] (Figure 3A). Other members of the *obscura* species group rarely display nuptial gift transfer [3,7]. When a male fly of *D. subobscura* finds a female, he extends a foreleg toward her to tap her abdomen (tapping), and then extends his proboscis toward her. The male may rapidly open and close two wings several times (scissoring) and slowly swing his bilateral midlegs (midleg swing). Each of these courtship acts is triggered by the distinct motion pattern of a visual target [73]. After repeating these displays, the male positions himself in front of the female with a quick motion, then fully opens both wings, culminating in nuptial gift transfer when the male is successful: the male extends his proboscis with a regurgitated droplet on its tip, i.e., the labellum, and the female responds to the gift by protruding her proboscis, so that the labellar surfaces of the male and female come into contact [3]. The two proboscises move back and forth while maintaining the labellar contact for seconds and the female swallows the droplet provided by the male. The male then quickly circles to the rear and attempts to mount the female. Despite the overall difference in mating behavior, the *fru* gene could play a master regulator role for courtship circuit formation in *D. subobscura*, as it does in *D. melanogaster*. Assuming that was the case, Tanaka [74] produced *fru* variants in *D. subobscura*, with the aid of CRISPR/Cas9 genome editing and *piggyBac*- and ø31-mediated transgenesis [75]. Venus-tagged channelrhodopsin was integrated into exon-2 of the *fru* locus, allowing visualization and artificial activation of the *fru*-labeled circuit in *D. subobscura*. The *fru*-labeled circuit thus visualized with Venus in *D. subobscura* was very similar to that in *D. melanogaster* at the gross anatomy level, and some of the key neural elements of the *fru*-labeled circuit, such as P1 and mAL neurons, were identifiable in *D. subobscura* by their characteristic structures common to the *D. melanogaster* counterparts [74]. Optogenetic activation of the *fru*-labeled circuit in *D. subobscura* induced robust behavioral responses such as abdominal bending, which resembles the motor act for copulation. Remarkably, in addition to the copulation-like abdominal bending, some male flies exhibited regurgitation, an essential component of nuptial gift transfer, upon *fru*-circuit activation [74]. These observations suggest that, although the *fru*-labeled circuit is conserved across species in terms of its overall anatomy and courtship-inducing ability, this circuit also triggers a species-specific courtship element, i.e., nuptial gift transfer. After eating to satiety, Dipterans often regurgitate and reingest their crop contents [76]. Regurgitation after feeding is suggested to be a strategy by which flies eliminate their crop water load to concentrate crop solute [77]. One interesting evolutionary scenario is that regurgitation as a component of feeding behavior may have been incorporated into the courtship ritual in *D. subobscura*. This idea further invites supposition that, during evolution, a neural module for feeding-associated regurgitation *in toto* was placed downstream of the courtship command system in an ancestral species of *D. subobscura* (Figure 3B,C). It might be that the *fru* gene of the *subobscura* ancestor happened to acquire *cis* elements that drive expression in neurons composing the regurgitation module and, as a consequence, nuptial gift transfer became possible as an innate repertoire of *D. subobscura* courtship behavior. Identification of *D. subobscura* neurons involved in nuptial gift transfer would be an important next step toward understanding of the neural bases for the evolution of novel behavioral traits. 

## 8. Perspectives

The current state of knowledge regarding the species differences in neural circuitry remains scanty, making it impossible to draw a generalized framework to illuminate causal links among the genes, circuits, and behaviors underlying the diversification of courtship rituals. Nonetheless, a few pioneering works have explored the neural underpinnings of behavioral divergence, and their findings augur explosive development in this field in the coming decade. In the very near future, exhaustive identification of neurons will be accomplished in *D. melanogaster*, accompanying databases for GAL4 lines that are highly specific for every neuron thus identified. Recent advances in single-cell or single-nucleus RNA sequencing techniques will open an avenue for deciphering the molecular landscape within seemingly similar Fru neurons across species. As we learned in the published works reviewed in this article, these GAL4s are easily introduced into non-model *Drosophila* species, where they can drive the expression of any gene of interest in homologous neurons in different species. This allows us to compare the structure and function of homologous neurons and the circuits they form across species, making it possible to determine which circuit element is different and how that change contributes to behavioral differences between species. Indeed, the CRISPR/Cas9 technique opened an avenue for the manipulation of any gene potentially involved in such behavioral diversity across species. The combinatorial uses of traditional genetics and state-of-the-art technologies in data science will provide a key to understanding how the neural circuit evolves to elaborate existing behavior and to create a novel behavioral repertoire.

## Figures and Tables

**Figure 1 genes-11-00157-f001:**
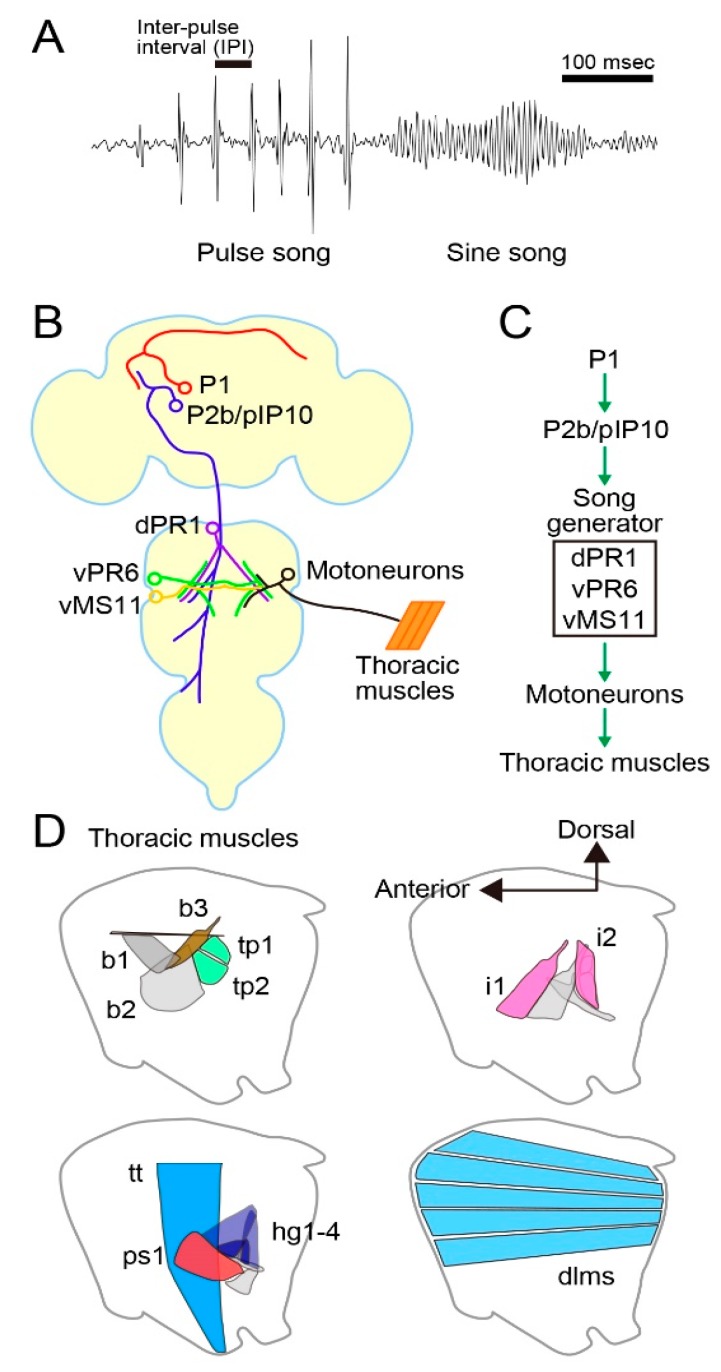
Neural and motor systems for courtship song generation in *Drosophila melanogaster*. (**A**) Song structure. (**B**, **C**) Neural components for song production (**B**) and their hierarchical organization (**C**). (**D**) Major muscles for song generation.

**Figure 2 genes-11-00157-f002:**
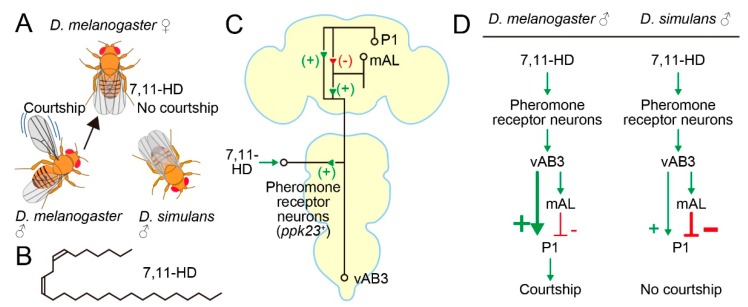
Circuit basis for the conspecific preference in mate choice. (**A**) *D. melanogaster* female-specific pheromone 7,11-HD attracts and repels male *D. melanogaster* and male *D. simulans*, respectively. (**B**) Structure of 7,11-HD. (**C**) A pathway through which the 7,11-HD information reaches to P1 neurons. + and – denote excitatory and inhibitory connections, respectively. (**D**) Proposed species differences in the pheromone processing pathway. Thick lines indicate predominant pathways.

**Figure 3 genes-11-00157-f003:**
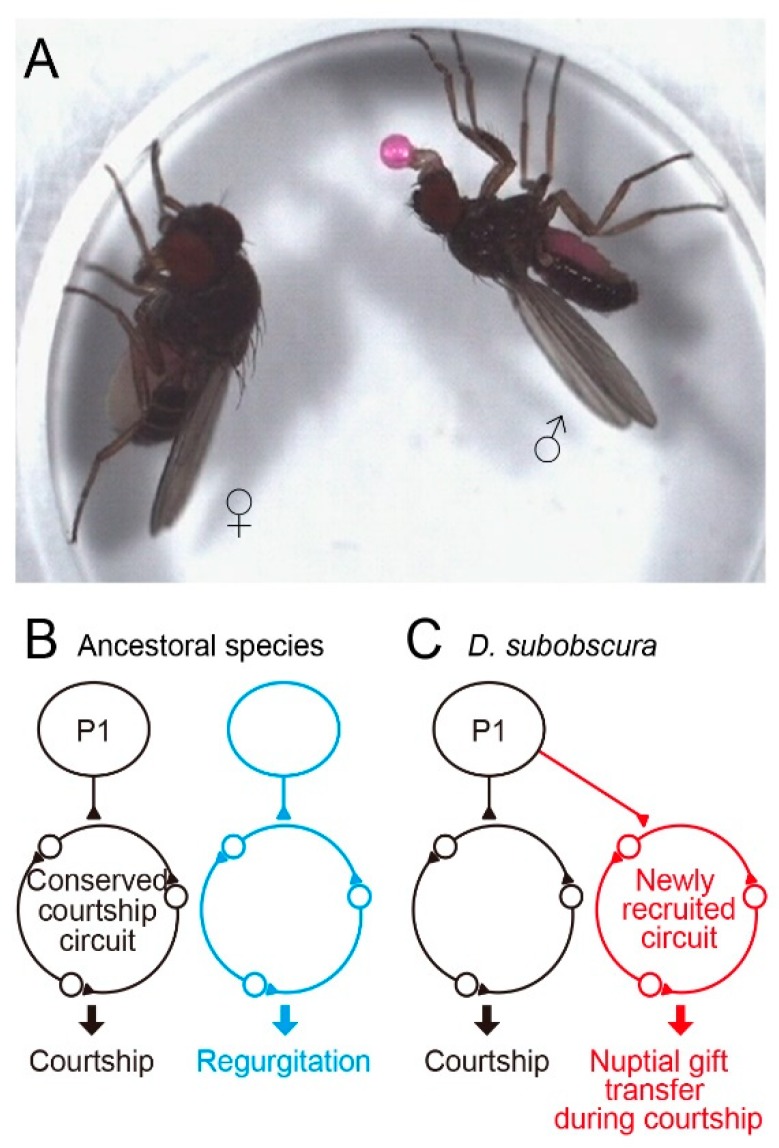
A possible circuit mechanism whereby *D. subobscura* acquired the novel courtship action element nuptial gift. (**A**) Nuptial gift transfer. (**B**) The regurgitation circuit operating independent of the courtship circuit in the ancestral state. (**C**) The regurgitation circuit neurons *in toto* happened to express the courtship master gene *fruitless*, resulting in the incorporation of this circuit into the courtship neural network in *D. subobscura*.

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
