# Peer review of "Behavioral Evolution of Drosophila: Unraveling the Circuit Basis"

_genes, 2020, doi:10.3390/genes11020157_

Round 1

Reviewer 1 Report

Dear Editor,

The review article by Sato et al is a well written, clear and succinct update on the evolution of mating behaviour circuitry in Drosophila. I believe it presents a nice contribution to Genes. I have a few comments whose consideration I believe would further improve the manuscript.

Comments:

Line 43: I’m tempted to disagree with the Authors statement that neuronal circuit evolution needs to precede phenotypic evolution in behaviour. Neuronal circuits are plastic and altered in both function and shape by experience (e.g. see review from Clayton et al, 2019, PNAS “Experience-dependent brain plasticity in persistence and change”. This means behaviour is restricted by a certain hard wiring of neuronal circuits, and this indeed dependent on neuronal circuitry, but it is also flexible within that circuitry. Sustained environmentally-induced changes in behaviour can induce the rewiring of synaptic connections changing neuronal circuitry. In this sense, behavioural phenotypes co-evolve with neuronal circuits, and it is more of an “chicken and egg” question. I would suggest the authors consider this.

Line 184: The statement “The cro mutation resides in the gene encoding the calmodulin-binding 185 transcription factor (Camta)…” is confusing to me. Line 176 describes cro as the gene croaker, while line line 184 implies cro is a mutation or allele in the gene Camta. Please clarify.

Line 205-211: While this is an interesting experiment, as stated above “the host D. melanogaster male which received a non-melanogaster fru locus continued to produce courtship song characteristic of D. melanogaster, regardless of whether he was a mutant for fru and/or dsx”, in the case of fru mutants, there would have been no endogenous fru to activate. So this theory would be precluded by the fru mutant experiment?

Perhaps the authors could incorporate how the findings of Ahmed et al (Evolution of Mechanisms that Control Mating in Drosophila Males, 2019, Cell Reports 27, 2527–2536) fit into our understanding of mate preference described in section 6.

Fig.1: Just a note that the included version of the figure has an unintended “Screenshot” label covering the muscles in bottom right.

Author Response

Point-by-point replies to the reviewers’ comments

Reviewer 1

Q1. Line 43

A1. Thank you for the interesting discussion. We think that experience-dependent behavioral changes the reviewer pointed out also rely on changes in a neural circuit. As a simple solution for revisions of this part, we deleted the entire sentence in question, i.e., “In other words, neural circuit evolution precedes phenotypic evolution in behavior”.

Q2.

A2. We rephrased the relevant passage as “The cro gene encodes the calmodulin-binding transcription factor (Camta) and, when mutated, displays multiple defects in mating behavior including prolongation of pulse song IPIs.” We hope that this revision made the statement straightforward.

Q3.

A3. In this passage, we intended to discuss the possibility that the heterologous species-specific song motor circuit transplanted from a donner species failed to receive the sensory-triggered command for singing, whereas the endogenous song motor circuit of the host species operated normally, presumably because FruM proteins were supplied to the conserved core neural elements (e.g., P1 neurons) via the heterologous promoter and thus high-fidelity neural activity required for song control was generated. Practically speaking, we thought that the difference in the methods, i.e., <forced CNS activation> vs. <natural (female-mediated) CNS activation via the PNS>, might be the cause of the different results. To make it easier for readers to follow the context, we rephrased “endogenous fru circuit” and “heterologous fru circuit” as “endogenous fru song motor circuit” and “heterologous fru motor circuit”, respectively.

Q4.

A4. We cited Ahmed et al. (2019) in section 6.

Q5.

A5. The “screenshot” label was introduced into Fig. 1 during the editorial process. Because this figure was also laterally expanded when pasted to the same word file, we replaced it with the figure with no distortion.

Reviewer 2 Report

The review by Sato et al., is a timely relevant manuscript providing an overview of recent studies that shed light on neuronal and molecular mechanisms that can explain the evolution of courtship related behaviors across Drosophila species. The review clearly describes the comprehensive dissection of circuits mediating different aspects of courtship behavior in Drosophila melanogaster as a unique starting point to study the evolution of behavior. The authors focus on 3 major species-specific differences: evolution of song, pheromone dependent mate choice preference and a unique nuptial gift behavior seen in Drosophila subobscura. The review presents and discusses attempts to identify variation in Fru circuitry with emphasis on differences in sensory mechanisms and motor output circuits, and genes that can modulate the physiology of these neurons as means that lead to behavioral species differences. Overall, I find this an important review that provides a conceptual framework for understanding neurogenetic mechanisms that shape the evolution of innate behaviors.

Comments:

The introduction section provides a general framework for the field of the evolution of species-specific behaviors its interplay with the evolution of morphological traits. I recommend adding some general examples for this before diving into Drosophila such as examples for the way by which molecular mechanisms shape the evolution of morphological changes are studied, and mention the study of other species-specific behaviors such as Konrad Lorenz studies, and the challenges of studying this at the neurogenetic level.

There is a lack of general overview of the possible circuit and molecular mechanisms that can be at the basis of behavior evolution. This kind of model can serve as a conceptual framework for the sections discussing song, mate choice, and nuptial gift behavior. It is worth mentioning that evolution can occur at the circuit wiring level or circuit physiology. This can be mediated by genes that can specify the development of species-specific variation in circuit anatomy, or genes that regulate the physiology of the neurons. In the molecular aspects, one should consider RNA editing, M6a RNA methylation, non-coding RNA, micro-peptides and so on as possible mechanisms that contribute to species-specific differences in the physiology and function of circuits. In addition, one should consider anatomical differences in muscles such as existing dimorphic differences, and differences in non-neuronal components of the nervous system such as the role of glia cells modulating the physiology of neurons and other non-neuronal components such as possible variation in the olfactory sensilla proteome (Obps) and even possible contribution of the microbiome.

At the perspectives section, it would be valuable to describe new advances in cell-type-specific and single-cell/nuclei RNAseq technologies that can be used as a comparative force for the molecular landscape within seemingly similar Fru neurons across species. Regarding the use of CRISPR/Cas9 approaches to investigate the function of genes, one should discuss the need to control such manipulations in a spatial and temporal manner to achieve mechanistic understanding.

The title is not clear: what do you mean by Big & Small in the behavioral evolution?

Author Response

Reviewer 2

Q6.

A6. Thank you very much for the constructive comments on our manuscript. According to the reviewer’s suggestion, we added, in the Introduction, a sentence to point out that Konrad Lorenz had an inspiring (and correct) idea that quantitative changes in gene expression levels could result in the presence of a particular behavioral act in one species and the absence of the same act in the other species, with the citation of Lorenz (1965).

Q7.

A7. We think that it would be premature to discuss possible molecular mechanisms underlying behavioral evolution, because our knowledge on this is still scanty. The reviewer also pointed out the importance of distinction between changes at the circuit level and changes in circuit physiology. To emphasize the dichotomy of these two, we modified the second paragraph of the Introduction so that the words “wiring” and “physiology” were inserted in the text: “Thus, any change in a behavioral pattern is a consequence of alterations in the structure (wiring) and/or function (physiology) of the neural circuit underlying the behavior.”

Q8.

A8. Thank you for the valuable comments. We agree with the reviewer’s view that single cell/nuclei RNAseq technologies will give us a solution for distinguishing molecular landscapes of individual fru neurons within a species and among homologous neurons across species. We expanded the perspective section to include these considerations.

Q9.

A9. To avoid any confusion, we deleted “Big & Small”, resulting in the new title, “Behavioral evolution of Drosophila: unraveling the circuit basis”. With the phrase “Big & Small”, we wanted to refer to large changes in behavioral evolution (macroevolution) and small changes in behavioral evolution (microevolution), while the phrase was taken from the title of a British children’s TV program.